# MATRIX MULTILAYER PERCEPTRON

## ABSTRACT

Models that output a vector of responses given some inputs, in the form of a conditional mean vector, are at the core of machine learning. This includes neural networks such as the multilayer perceptron (MLP). However, models that output a symmetric positive definite (SPD) matrix of responses given inputs, in the form of a conditional covariance function, are far less studied, especially within the context of neural networks. Here, we introduce a new variant of the MLP, referred to as the matrix MLP, that is specialized at learning SPD matrices. Our construction not only respects the SPD constraint, but also makes explicit use of it. This translates into a model which effectively performs the task of SPD matrix learning even in scenarios where data are scarce. We present an application of the model in heteroscedastic multivariate regression, including convincing performance on six real-world datasets.

## 1 INTRODUCTION

For certain applications, it is desirable to construct a conditional covariance matrix as a function of the input (the explanatory variable). The problem arises, for instance, in spatial (and spatio-temporal) statistics, in relation to the heteroscedastic multivariate regression (e.g., Pourahmadi, 1999; Hoff & Niu, 2012; Fox & Dunson, 2015), where we deal with multivariate response measurements for which the typical assumption of homoscedasticity may not be suitable. In such cases, we require models that estimate the covariance matrix that captures the spatial variations in correlations between the elements of the response vector. The covariance matrix is a symmetric positive definite (SPD) matrix which can be challenging to estimate due to its non-Euclidean geometry (Pennec et al., 2005). The central problem that this work is concerned with is learning SPD matrices using neural networks.

To motivate our discussion, consider solving the problem of SPD matrix learning using a multilayer perceptron (MLP) as an example of a fully connected neural network (e.g., Goodfellow et al., 2016). To meet the SPD constraint, one would need to tailor the output layer of the MLP so that the estimated covariance matrix satisfies the SPD requirement. One possible approach would be to use the Cholesky decomposition. The main concern is that this approach does not take into account the non-Euclidean geometry of the SPD matrices. Using empirical evaluations, we will show that the use of "wrong geometry" results in the poor estimation of the SPD matrices in particular where data are scarce.

The primary objective here is to design a nonlinear architecture using neural networks that can effectively perform the task of SPD matrix learning. More specifically, our *main contribution* is to show how to alter the architecture of the MLP in such a way that it not only respects the SPD constraint, but also makes an explicit use of it. We will achieve this by: 1) Explicitly taking the non-Euclidean geometry of the underlying SPD manifolds (e.g., Pennec et al., 2005) into account by designing a new loss function, and 2) by deriving a new backpropagation algorithm (Rumelhart et al., 1986) that respects the SPD nature of the matrices. This new model will be referred to as the *matrix multilayer perceptron (mMLP)*[1]. The mMLP makes use of positive-definite kernels to satisfy the SPD requirement across all layers. Hence, it provides a natural way of enabling deep SPD matrix learning.

We take a step-by-step approach in the development of the model. We first develop a simplified version of the resulting model that is designed for learning SPD matrices (Section 3). We then extend this model into its most general form that can be used for joint estimation of the conditional mean function and the conditional covariance function in a mean-covariance regression setting (Section 3.2). An application of the model is discussed in the context of the heteroscedastic multivariate regression.

---

[1]An implementation of this work will be made available via GitHub.

## 2 RELATED WORK

**SPD manifold metric.** Earlier approaches for analyzing SPD matrices relied on the Euclidean space. But over the past decade, several studies suggest that non-Euclidean geometries such as the Riemannian structure may be better suited (e.g., Arsigny et al., 2006; Pennec et al., 2005). In this work, we consider the von Neumann divergence (e.g., Nielsen & Chuang, 2000) as our choice of the SPD manifold metric which is related to the Riemannian geometry. Previously, Tsuda et al. (2005) used this divergence in derivation of the matrix exponentiated gradients. Their work suggests its effectiveness for measuring dissimilarities between positive definite (PD) matrices.

**SPD manifold learning.** There are multiple approaches towards the SPD matrix learning, via the flattening of SPD manifolds through tangent space approximations (e.g., Oncel Tuzel, 2008; Fathy et al., 2016), mapping them into reproducing kernel Hilbert spaces (Harandi et al., 2012; Minh et al., 2014), or geometry-aware SPD matrix learning (Harandi et al., 2014). While these methods typically employ shallow learning, the more recent line of research aims to design a deep architecture to nonlinearly learn target SPD matrices (Ionescu et al., 2015; Huang & Gool, 2017; Masci et al., 2015; Huang et al., 2018). Our method falls in this category but differs in the problem formulation. While the previous methods address the problem where the input is an SPD matrix and the output is a vector, we consider the reverse problem where the input is a matrix with an arbitrary size and the output is an SPD matrix.

**Backpropagation.** Our extension of the matrix backpropagation differs from the one introduced by Ionescu et al. (2015). In their work, the necessary partial derivatives are computed using a two-step procedure consisting of first computing the functional that describes the variations of the upper layer variables with respect to the variations of the lower layer variables, and then computing the partial derivatives with respect to the lower layer variables using properties of the matrix inner product. In contrast, we make use of the concept of $\alpha$-derivatives (Magnus, 2010) and its favorable generalization properties to derive a procedure which *closely* mimics the standard backpropagation.

## 3 MATRIX MULTILAYER PERCEPTRON

**Preliminaries: Matrix $\alpha$-derivative.** Throughout this work we adopt the *narrow* definition of the matrix derivatives known as the $\alpha$-derivative (Magnus, 2010) in favor of the broad definition, the $\omega$-derivative. The reason for this is that the $\alpha$-derivative has better generalization properties. This choice turned out to be crucial in the derivation of the mMLP's backpropagation routine which involves derivatives of matrix functions w.r.t. the matrix of variables.

***Definition:*** Let $\mathbf{F}$ be an $m \times n$ matrix function of an $n \times q$ matrix of variables $\mathbf{X}$. The $\alpha$-derivative of $\mathbf{F}(\mathbf{X})$ is defined as (Magnus, 2010, Definition 2)

$$\mathrm{D}_{\mathbf{X}}\mathbf{F} := \frac{\partial\,\mathrm{vec}\mathbf{F}(\mathbf{X})}{\partial\,(\mathrm{vec}\mathbf{X})^{\top}}, \tag{1}$$

where $\mathrm{D}_{\mathbf{X}}\mathbf{F}$ is an $mp \times nq$ matrix which contains all the partial derivatives such that each row contains the partial derivatives of one function with respect to all variables, and each column contains the partial derivatives of all functions with respect to one variable.

For convenience, the $\alpha$-derivative' basic properties, including the product rule and the chain rule, are summarized in Appendix B.

### 3.1 THE BASIC FORM OF THE MMLP

**Activation matrix function.** Let $\mathbf{Z} = (\mathbf{z}_1, \ldots, \mathbf{z}_d)$ denote a matrix of variables $\mathbf{z}_i \in \mathbb{R}^d$. The activation function $\mathcal{K}(\mathbf{Z})$ defines a matrix function in the form of $[\mathcal{K}(\mathbf{Z})]_{i,j} = \kappa(\mathbf{z}_i, \mathbf{z}_j)$, $\forall i,j \in \{1, \ldots, d\}$, where $\kappa$ is some differentiable activation function outputting scalar values. In the following, we restrict ourselves to the kernel functions which form PD activation matrix functions. For numerical stability reasons, irrespective of the functional form of $\kappa$, we normalize the resulting matrix. This can be achieved by enforcing the trace-one constraint, $\mathcal{H}(\mathbf{Z}) = \mathcal{K}(\mathbf{Z})/\mathrm{tr}(\mathcal{K}(\mathbf{Z}))$, where $\mathcal{H}$ denotes a differentiable SPD activation matrix function of trace one. Without loss of generality, throughout this work, we use the Mercer sigmoid kernel (Carrington et al., 2014) defined as

$$\kappa(\mathbf{z}_i, \mathbf{z}_j) = \tanh(\alpha\mathbf{z}_i + \beta) \odot \tanh(\alpha\mathbf{z}_j + \beta), \tag{2}$$

where $\alpha$ and $\beta$ denote the slope and the intercept, respectively. Furthermore, $\odot$ denotes the dot product. In all experiments, we use default values of $\alpha = 1$ and $\beta = 0$. The The $\alpha$-derivative of the Mercer sigmoid kernel is computed in Appendix E.

**Model construction.** Let $\mathbf{X} \in \mathbb{R}^{p_1 \times p_2}$ indicate the input matrix and $\mathbf{Y} \in \mathbb{R}^{d_0 \times d_0}$ indicate the corresponding output matrix, an SPD matrix of trace one. The mMLP of $L$ hidden layers is shown as mMLP$: \mathbf{X} \to \widehat{\mathbf{Y}}$ and constructed as

$$\begin{cases} \widehat{\mathbf{Y}} = \mathcal{H}(\mathbf{Z}_0), & \mathbf{Z}_0 = \mathbf{W}_0 \mathbf{H}_1 \mathbf{W}_0^\top + \mathbf{B}_0, \\ \mathbf{H}_l = \mathcal{H}(\mathbf{Z}_l), & \mathbf{Z}_l = \mathbf{W}_l \mathbf{H}_{l+1} \mathbf{W}_l^\top + \mathbf{B}_l, \quad \forall\, 1 \le l \le L, \\ \mathbf{H}_{L+1} = \mathcal{H}(\mathbf{Z}_{L+1}), & \mathbf{Z}_{L+1} = \mathbf{W}_{L+1} \mathrm{vec}\mathbf{X}(\mathbf{W}_{L+1} \mathbf{1}_{p_1 p_2})^\top + \mathbf{B}_{L+1}. \end{cases} \quad (3)$$

The pair of $\mathbf{W}_l \in \mathbb{R}^{d_l \times d_{l+1}}, \forall 0 \le l \le L$, and $\mathbf{W}_{L+1} \in \mathbb{R}^{d_{L+1} \times p_1 p_2}$ are the weight matrices, $\mathbf{B}_l \in \mathbb{R}^{d_l \times d_l}, \forall 0 \le l \le L+1$, are the bias matrices, $\mathbf{Z}_l \in \mathbb{R}^{d_l \times d_l}, \forall 0 \le l \le L+1$, are the latent input matrices, and $\mathbf{H}_l \in \mathbb{R}^{d_l \times d_l}, \forall 1 \le l \le L+1$, are latent output SPD matrices of trace one.

**Design choice.** In the construction of (3), we have ensured that $\mathbf{H}_l$ are SPD matrices of trace one *across all layers* as opposed to only at the output layer. The idea behind this design choice is to propagate the nonlinearities introduced via the SPD activation matrix functions through all layers. This design choice turned out to be more effective than the alternative, and arguably simpler, design where the SPD requirement is met only at the output layer. We will discuss this further in Section 5.1.3, where we also present an illustrative numerical example.

**Loss function.** We consider the normalized von Neumann divergence (e.g., Nielsen & Chuang, 2000), also commonly known as the quantum relative entropy (QRE), as the base for the loss function. For two arbitrary SPD matrices of trace one, $\mathbf{\Phi}$ and $\widetilde{\mathbf{\Phi}}$, the normalized von Neumann divergence is defined as:

$$\Delta_{\mathrm{QRE}}(\widetilde{\mathbf{\Phi}} || \mathbf{\Phi}) = \mathrm{tr}(\widetilde{\mathbf{\Phi}} \mathfrak{log} \widetilde{\mathbf{\Phi}} - \widetilde{\mathbf{\Phi}} \mathfrak{log} \mathbf{\Phi}), \quad (4)$$

where $\mathfrak{log}$ denotes the matrix logarithm (taking $\mathbf{\Phi}$ as an example, it is computed using $\mathfrak{log}\mathbf{\Phi} = \mathbf{V} \mathrm{diag}(\log\boldsymbol{\lambda}) \mathbf{V}^\top$, where $\mathbf{V}$ and $\boldsymbol{\lambda}$ are the matrix of eigenvectors and the vector of eigenvalues from the eigendecomposition of $\mathbf{\Phi}$). The von Neumann divergence is asymmetric. However, it can be symmetrized by using the fact that the von Neumann entropy of trace one follows the class of generalized quadratic distances (Nielsen & Nock, 2007). Hence, we define the loss function as[2]

$$\ell_{\mathrm{QRE}}(\widehat{\mathbf{Y}}, \mathbf{Y}) = \frac{1}{2}(\Delta_{\mathrm{QRE}}(\widehat{\mathbf{Y}} || \mathbf{Y}) + \Delta_{\mathrm{QRE}}(\mathbf{Y} || \widehat{\mathbf{Y}})). \quad (5)$$

Taking $\alpha$-derivative of $\ell_{\mathrm{QRE}}$ involves taking partial derivatives through the eigendecomposition. In the following, we derive a method for analytically computing the derivative of $\ell_{\mathrm{QRE}}$.

**The $\alpha$-derivative of the symmetrized von Neumann divergence.** For the loss function defined in (5), using the $\alpha$-derivative's product rule and chain rule (refer to Appendix B), we obtain

$$\mathsf{D}_{\widehat{\mathbf{Y}}} \ell = \frac{1}{2} \mathsf{D}_{\widehat{\mathbf{Y}}} \mathrm{tr}((\widehat{\mathbf{Y}} - \mathbf{Y}) \mathfrak{log} \widehat{\mathbf{Y}}) - \frac{1}{2} \mathsf{D}_{\widehat{\mathbf{Y}}} \mathrm{tr}(\widehat{\mathbf{Y}} \mathfrak{log} \mathbf{Y}), \quad (6)$$

where the above two terms are computed using

$$\mathsf{D}_{\widehat{\mathbf{Y}}}(\mathrm{tr}((\widehat{\mathbf{Y}} - \mathbf{Y}) \mathfrak{log} \widehat{\mathbf{Y}})) = \underbrace{(\mathrm{vec}(\mathfrak{log}\widehat{\mathbf{Y}})^\top)^\top}_{1 \times d_0^2} + \underbrace{\begin{pmatrix} \mathrm{vec}(\widehat{\mathbf{Y}}^\top - \mathbf{Y}^\top) \odot \mathrm{vec}(\frac{\partial}{\partial \widehat{Y}_{11}} \mathfrak{log}\widehat{\mathbf{Y}}) \\ \mathrm{vec}(\widehat{\mathbf{Y}}^\top - \mathbf{Y}^\top) \odot \mathrm{vec}(\frac{\partial}{\partial \widehat{Y}_{21}} \mathfrak{log}\widehat{\mathbf{Y}}) \\ \vdots \\ \mathrm{vec}(\widehat{\mathbf{Y}}^\top - \mathbf{Y}^\top) \odot \mathrm{vec}(\frac{\partial}{\partial \widehat{Y}_{d_0 d_0}} \mathfrak{log}\widehat{\mathbf{Y}}) \end{pmatrix}^\top}_{1 \times d_0^2}, \quad (7)$$

$$\mathsf{D}_{\widehat{\mathbf{Y}}}(\mathrm{tr}(\widehat{\mathbf{Y}} \mathfrak{log} \mathbf{Y})) = \underbrace{(\mathrm{vec}(\mathfrak{log}\mathbf{Y})^\top)^\top}_{1 \times d_0^2}. \quad (8)$$

---

[2]Note that there are multiple ways of symmetrizing the von Neumann divergence. Our choice in (5) resembles to the $\alpha$-Jensen–Shannon divergence for $\alpha = 1$ (Nielsen, 2010).

The remaining part in the computation of (6) is to evaluate $\frac{\partial}{\partial \widehat{Y}_{ij}} \log \widehat{\mathbf{Y}}$, for all $i, j \in \{1, \ldots, d_0\}$, which involves taking derivatives through the eigendecomposition. In the following, we take a similar approach as in Papadopoulo & Lourakis (2000) to compute the necessary partial derivatives.

Let $\widehat{\mathbf{Y}} = \mathbf{\Upsilon} \mathrm{diag}(\lambda_1, \ldots, \lambda_{d_0}) \mathbf{\Upsilon}^\top$ be the eigendecomposition. We can write

$$
\begin{aligned}
\frac{\partial}{\partial \widehat{Y}_{ij}} \log \widehat{\mathbf{Y}} &= \frac{\partial}{\partial \widehat{Y}_{ij}} \mathbf{\Upsilon} \mathbf{\Lambda} \mathbf{\Upsilon}^\top, \quad \text{where} \quad \mathbf{\Lambda} = \mathrm{diag}(\log \lambda_1, \ldots, \log \lambda_{d_0}), \\
&= \frac{\partial \mathbf{\Upsilon}}{\partial \widehat{Y}_{ij}} \mathbf{\Lambda} \mathbf{\Upsilon}^\top + \mathbf{\Upsilon} \frac{\partial \mathbf{\Lambda}}{\partial \widehat{Y}_{ij}} \mathbf{\Upsilon}^\top + \mathbf{\Upsilon} \mathbf{\Lambda} \frac{\partial \mathbf{\Upsilon}^\top}{\partial \widehat{Y}_{ij}}.
\end{aligned}
\tag{9}
$$

By multiplying (9) from left and right by $\mathbf{\Upsilon}^\top$ and $\mathbf{\Upsilon}$ respectively, we obtain:

$$
\mathbf{\Upsilon}^\top \frac{\partial}{\partial \widehat{Y}_{ij}} \log \widehat{\mathbf{Y}} \, \mathbf{\Upsilon} = \mathbf{\Upsilon}^\top \frac{\partial \mathbf{\Upsilon}}{\partial \widehat{Y}_{ij}} \mathbf{\Lambda} + \frac{\partial \mathbf{\Lambda}}{\partial \widehat{Y}_{ij}} + \mathbf{\Lambda} \frac{\partial \mathbf{\Upsilon}^\top}{\partial \widehat{Y}_{ij}} \mathbf{\Upsilon} = \mathbf{\Xi}_{ij}(\mathbf{\Upsilon}) \mathbf{\Lambda} + \frac{\partial \mathbf{\Lambda}}{\partial \widehat{Y}_{ij}} - \mathbf{\Lambda} \mathbf{\Xi}_{ij}(\mathbf{\Upsilon}), \tag{10}
$$

where we have defined $\mathbf{\Xi}_{ij}(\mathbf{\Upsilon}) = \mathbf{\Upsilon}^\top \frac{\partial}{\partial \widehat{Y}_{ij}} \mathbf{\Upsilon}$ and used the fact that $\mathbf{\Xi}_{ij}(\mathbf{\Upsilon})$ is an antisymmetric matrix, $\mathbf{\Xi}_{ij}(\mathbf{\Upsilon}) + \mathbf{\Xi}_{ij}^\top(\mathbf{\Upsilon}) = \mathbf{0}$, which in turn follows from the fact that $\mathbf{\Upsilon}$ is an orthonormal matrix,

$$
\mathbf{\Upsilon}^\top \mathbf{\Upsilon} = \mathbf{I}_{d_0} \Rightarrow \frac{\partial \mathbf{\Upsilon}^\top}{\partial \widehat{Y}_{ij}} \mathbf{\Upsilon} + \mathbf{\Upsilon}^\top \frac{\partial \mathbf{\Upsilon}}{\partial \widehat{Y}_{ij}} = \mathbf{\Xi}_{ij}^\top(\mathbf{\Upsilon}) + \mathbf{\Xi}_{ij}(\mathbf{\Upsilon}) = \mathbf{0}. \tag{11}
$$

Taking the antisymmetric property of $\mathbf{\Xi}_{ij}(\mathbf{\Upsilon})$ into account in (10), we obtain

$$
\frac{\partial}{\partial \widehat{Y}_{ij}} \log \lambda_k = \Upsilon_{ik} \Upsilon_{jk}, \tag{12}
$$

$$
\mathbf{\Xi}_{ij}(\Upsilon_{kl}) = \frac{\Upsilon_{ik} \Upsilon_{jl} + \Upsilon_{il} \Upsilon_{jk}}{2(\log \lambda_l - \log \lambda_k)}, \quad \forall l \neq k. \tag{13}
$$

It is notable that by construction, we do not have repeating eigenvalues, that is $\lambda_k \neq \lambda_l, \ \forall k \neq l$, so there exists a unique solution to (13). Once $\mathbf{\Xi}_{ij}(\mathbf{\Upsilon})$ is computed, it follows that

$$
\frac{\partial \mathbf{\Upsilon}}{\partial \widehat{Y}_{ij}} = \mathbf{\Upsilon} \mathbf{\Xi}_{ij}(\mathbf{\Upsilon}), \qquad \frac{\partial \mathbf{\Upsilon}^\top}{\partial \widehat{Y}_{ij}} = -\mathbf{\Xi}_{ij}(\mathbf{\Upsilon}) \mathbf{\Upsilon}^\top. \tag{14}
$$

In summary, the necessary partial derivatives for computing (9) is given by (12) and (14). Once (9) is computed for all $i, j$, we can evaluate (7) and ultimately evalaute (6).

**Optimization.** The remaining steps are feed-forward computation, backpropagation, and learning, as in the standard MLP. However, here, the backpropagation requires taking derivatives with respect to the matrix functions. These steps are described in Appendix C.

### 3.2 THE GENERAL FORM OF THE mMLP

We now discuss a general version of the mMLP which produces both a vector and an SPD matrix as outputs. An important application of this model is in heteroscedastic multivariate regression which we will discuss in Section 5.2.

**Model construction.** As before, let $\mathbf{X} \in \mathbb{R}^{p_1 \times p_2}$ denote the input matrix. The corresponding outputs in this case are: an SPD matrix of trace one $\mathbf{Y} \in \mathbb{R}^{d_0 \times d_0}$ and $\mathbf{y} \in \mathbb{R}^{r_0}$. The mMLP of $L$ hidden layers is denoted by $\mathrm{mMLP} : \mathbf{X} \to \{\widehat{\mathbf{y}}, \widehat{\mathbf{Y}}\}$ and constructed as:

$$
\begin{cases}
\widehat{\mathbf{y}} = \mathfrak{h}(\mathbf{z}_0), & \mathbf{z}_0 = \mathbf{C}_0 \widehat{\mathbf{Y}} \mathbf{A}_0 \mathbf{h}_1 + \mathbf{b}_0, \\
\widehat{\mathbf{Y}} = \mathcal{H}(\mathbf{Z}_0), & \mathbf{Z}_0 = \mathbf{W}_0 \mathbf{H}_1 \mathbf{W}_0^\top + \mathbf{B}_0, \\
\mathbf{h}_l = \mathfrak{h}(\mathbf{z}_l), & \mathbf{z}_l = \mathbf{C}_l \mathbf{H}_l \mathbf{A}_l \mathbf{h}_{l+1} + \mathbf{b}_l, \quad \forall 1 \leq l \leq L, \\
\mathbf{H}_l = \mathcal{H}(\mathbf{Z}_l), & \mathbf{Z}_l = \mathbf{W}_l \mathbf{H}_{l+1} \mathbf{W}_l^\top + \mathbf{B}_l, \quad \forall 1 \leq l \leq L, \\
\mathbf{h}_{L+1} = \mathfrak{h}(\mathbf{z}_{L+1}), & \mathbf{z}_{L+1} = \mathbf{C}_{L+1} \mathbf{H}_{L+1} \mathbf{A}_{L+1} \mathbf{1} + \mathbf{b}_{L+1}, \\
\mathbf{H}_{L+1} = \mathcal{H}(\mathbf{Z}_{L+1}), & \mathbf{Z}_{L+1} = \mathbf{W}_{L+1} \mathrm{vec} \mathbf{X}(\mathbf{W}_{L+1} \mathbf{1}_{p_1 p_2})^\top + \mathbf{B}_{L+1},
\end{cases}
\tag{15}
$$

where $\mathbf{h}_l \in \mathbb{R}^{r_l}, \mathbf{H}_l \in \mathbb{R}^{d_l \times d_l}, \forall 1 \leq l \leq L+1$, $\mathbf{z}_l, \mathbf{b}_l \in \mathbb{R}^{r_l}$, $\mathbf{Z}_l, \mathbf{B}_l \in \mathbb{R}^{d_l \times d_l}$, $\mathbf{C}_l \in \mathbb{R}^{r_l \times d_l}, \forall 0 \leq l \leq L+1$, $\mathbf{A}_l \in \mathbb{R}^{d_l \times r_{l+1}}$, and $\mathbf{W}_l \in \mathbb{R}^{d_l \times d_{l+1}}, \ \forall 0 \leq l \leq L$. Just as in the standard MLP, $\mathfrak{h}$ is an activation function of choice, e.g., the hyperbolic tangent function.

**Loss function.** The loss function here needs to be designed with the specific application in mind. In the case of the heteroscedastic multivariate regression, the loss can be defined based on the log-likelihood. This will be discussed in Section 4.

**Optimization.** The remaining steps of feed-forward computation, backpropagation, and learning, are all described in Appendix D.

## 4 THE MMLP IN HETEROSCEDASTIC MULTIVARIATE REGRESSION

In this section, we discuss an application of the mMLP in relation to the heteroscedastic multivariate regression, more specifically, the joint mean-covariance regression task.

**Model construction.** Let $\mathcal{D}_{\mathrm{train}} = \{\mathbf{y}^{(i)}, \mathbf{x}^{(i)}\}_{i=1}^{n}$ be our training dataset consisting of a set of inputs $\mathbf{x}_i \in \mathbb{R}^{d_\mathbf{x}}$ and a set of responses $\mathbf{y}_i \in \mathbb{R}^{d_\mathbf{y}}$. Consider the following multivariate regression problem: $\mathbf{y}^{(i)} = \boldsymbol{f}(\mathbf{x}^{(i)}) + \mathbf{e}^{(i)}$, where $\boldsymbol{f}$ is a nonlinear function, and $\mathbf{e}^{(i)}$ is the additive noise on the $i$-th response measurement. The goal is estimation of the conditional mean function $\mathsf{E}[\mathbf{y}\,|\,\mathbf{x}_*]$ and the conditional covariance function $\mathsf{Var}[\mathbf{y}\,|\,\mathbf{x}_*]$ for an unseen input $\mathbf{x}_* \in \mathcal{D}_{\mathrm{test}}$.

We consider two likelihood models based on our choices of the noise model, namely, the multivariate Gaussian model and its generalization the multivariate power exponential model.

*Multivariate Gaussian model.* We first consider the noise model to follow a zero-mean multivariate Gaussian (mG) distribution with a dense covariance matrix, that is $\mathbf{e}_i \sim \mathcal{N}(\mathbf{0}, \boldsymbol{\Sigma}_i)$. Let $\boldsymbol{\Sigma}_i = \eta_i \boldsymbol{\Omega}_i$, where $\eta_i = \mathrm{tr}(\boldsymbol{\Sigma}_i)$ and $\boldsymbol{\Omega}_i$ is a trace-one matrix. The noise model can accordingly be reformulated as $\mathbf{e}_i \sim \mathcal{N}_{\mathrm{tr1}}(\mathbf{0}, \boldsymbol{\Omega}_i, \eta_i)$ where $\mathcal{N}_{\mathrm{tr1}}$ is referred to as the *trace-one mG distribution*[3]. Although these two formulations of the mG distribution are identical, we find it easier to work with the latter. This is because the output layer of the mMLP model in (15) operates under the trace-one constraint. It is of course possible to drop the trace-one constraint from the output layer, but we would then also be forced to use a different kernel function than the Mercer sigmoid kernel in that layer. Instead, we find it easier to work with this reformulated mG distribution with a trace-one covariance matrix which allows us to use the same choice of kernel function (2) across all layers.

Given $\mathbf{e}_i \sim \mathcal{N}(\mathbf{0}, \boldsymbol{\Omega}_i, \eta_i)$, the likelihood is defined as

$$\ell_{\mathcal{N}_{\mathrm{tr1}}} := \log \mathcal{N}_{\mathrm{tr1}}(\mathbf{y}; \boldsymbol{\mu}, \boldsymbol{\Omega}, \eta), \quad \mathrm{mMLP}_{\boldsymbol{\theta}} : \mathbf{x} \rightarrow \{\boldsymbol{\mu}, \boldsymbol{\Omega}, \eta\}, \tag{16}$$

where $\mathsf{E}[\mathbf{y}\,|\,\mathbf{x}] = \boldsymbol{\mu}$, and $\mathsf{Var}[\mathbf{y}\,|\,\mathbf{x}] = \eta \boldsymbol{\Omega}$. The set $\boldsymbol{\theta}$ includes all the neural network parameters.

*Multivariate power exponential model.* We next consider the noise model to follow a zero-mean multivariate power exponential (mPE) distribution (e.g., Gómez et al., 1998) which is a generalized variant of the mG distribution, that is $\mathbf{e}^{(i)} \sim \mathcal{E}(\mathbf{0}, \boldsymbol{\Sigma}_i, \alpha_i, \beta_i)$ where $\boldsymbol{\Sigma}_i$ is a symmetric real dispersion matrix and the pair of $\alpha_i \in \mathbb{R}^+$ and $\beta_i \in \mathbb{R}^+$ control the tail and the shape of the distribution. As a special case, for $\alpha = 1$ and $\beta = 1$, the mPE includes the mG distribution[4].

As in the case of the mG distribution, we find it easier to work with a reformulated variant where the dispersion matrix is of trace-one. Let $\boldsymbol{\Sigma}_i = \eta_i \boldsymbol{\Omega}_i$, where $\eta_i = \mathrm{tr}(\boldsymbol{\Sigma}_i)$. The noise model can accordingly be represented as $\mathbf{e}^{(i)} \sim \mathcal{E}_{\mathrm{tr1}}(\mathbf{0}, \boldsymbol{\Omega}_i, \alpha_i, \beta_i, \eta_i)$ where $\mathcal{E}_{\mathrm{tr1}}$ is referred to as the *trace-one mPE distribution*[5].

Given $\mathbf{e}^{(i)} \sim \mathcal{E}_{\mathrm{tr1}}(\mathbf{0}, \boldsymbol{\Omega}_i, \alpha_i, \beta_i, \eta_i)$, the likelihood is defined as:

$$\ell_{\mathcal{E}_{\mathrm{tr1}}} := \log \mathcal{E}_{\mathrm{tr1}}(\mathbf{y}; \boldsymbol{\mu}, \boldsymbol{\Omega}, \alpha, \beta, \eta), \quad \mathrm{mMLP}_{\boldsymbol{\theta}} : \mathbf{x} \rightarrow \{\boldsymbol{\mu}, \boldsymbol{\Omega}, \alpha, \beta, \eta\}, \tag{17}$$

where $\mathsf{E}[\mathbf{y}\,|\,\mathbf{x}] = \boldsymbol{\mu}$ and $\mathsf{Var}[\mathbf{y}\,|\,\mathbf{x}] = \eta \alpha \nu(\beta) \boldsymbol{\Omega}$, where $\nu(\beta) = \frac{2^{1/\beta} \Gamma\left(\frac{d+1}{2\beta}\right)}{d\,\Gamma\left(\frac{d}{2\beta}\right)}$. The set $\boldsymbol{\theta}$ includes all the neural network parameters.

**Optimization.** Finally, the optimization involves learning the neural network parameters $\boldsymbol{\theta}$ by maximizing the likelihoods given by (16) and (17).

---

[3] Refer to Appendix F.1 for the exact functional form of the trace-one mG distribution.

[4] Figure G.1.1 visualizes the probability density function of the distribution for selected values of $\alpha$ and $\beta$.

[5] Refer to Appendix G for the exact functional form of the trace-one mPE distribution and its basic properties.

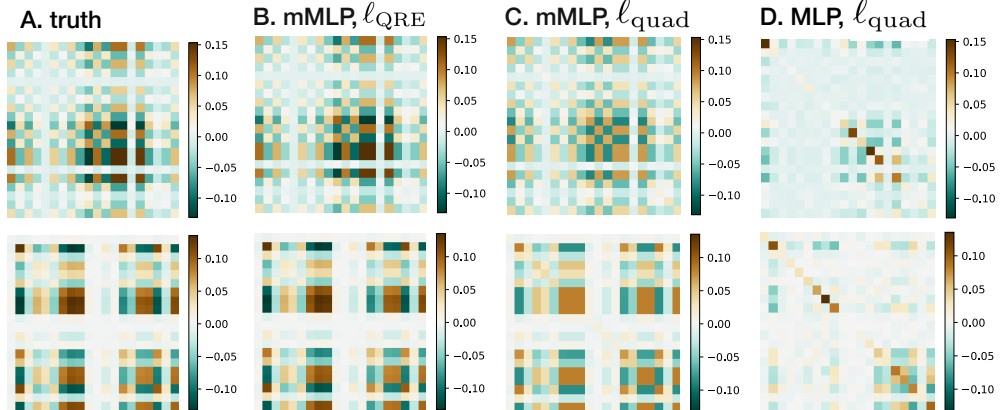

Figure 1: SPD matrix learning (refer to Example 1 in Section 5.1.1). Two instances of target covariance (SPD) matrices ($20 \times 20$). (B, C) Estimated covariance matrices by the mMLP using $\ell_{\mathrm{QRE}}$, and using $\ell_{\mathrm{quad}}$. (D) Estimated covariance matrices using the Cholesky-based MLP and $\ell_{\mathrm{quad}}$.

## 5 EXPERIMENTS

Experiments are divided into two parts. The first part is on the empirical validation of the mMLP model in a supervised task of learning SPD matrices using synthetic data. The second part discusses our results in heteroscedastic multivariate regression on real-world datasets.

### 5.1 SPD MATRIX LEARNING

#### 5.1.1 THE CHOICE OF LOSS FUNCTION

Consider the problem of learning SPD matrices on synthetic data using the mMLP model of (3). The objectives are to validate the model, and to evaluate the effect of the choice of the loss function on the overall performance.

The following loss functions are considered for this analysis: The first candidate is the loss function based on the normalized von Neumann divergence $\ell_{\mathrm{QRE}}(\widehat{\mathbf{Y}}, \mathbf{Y})$ given by (5). The $\ell_{\mathrm{QRE}}$ is related to the Riemannian geometry. The second candidate is the quadratic loss, $\ell_{\mathrm{quad}}(\widehat{\mathbf{Y}}, \mathbf{Y}) = \mathrm{tr}((\widehat{\mathbf{Y}} - \mathbf{Y})(\widehat{\mathbf{Y}} - \mathbf{Y})^{\top})$, which is related to the Euclidean geometry.

**Example 1.** Consider the set $\mathcal{D}_{\mathrm{train}} = \{\mathbf{x}_i, \mathbf{Y}_i\}_{i=1}^{n_{\mathrm{train}}}$ of inputs $\mathbf{x}_i \in \mathbb{R}^{20}$ and corresponding SPD matrix outputs $\mathbf{Y}_i \in \mathbb{R}^{d_0 \times d_0}$ which are in this case dense covariance matrices (refer to Appendix H.1 for details on the data generation). The goal is to estimate the covariance matrices $\widehat{\mathbf{Y}}$ associated to the input vectors from the unseen test set $\mathcal{D}_{\mathrm{test}} = \{\mathbf{x}_i\}_{i=1}^{n_{\mathrm{test}}}$. The training size is varied between $n_{\mathrm{train}} = \{10^2, 10^4\}$ samples. The analysis is carried out for $d_0 = \{10, 20\}$. Two examples of the test target outputs $\mathbf{Y}_i$ for $d_0 = 20$ and $n_{\mathrm{train}} = 10^2$ are visualized in Figure 1-A.

The mMLP models (3) are trained using 3 layers (20 units per layer, $d_l = 20$) under our two choices of loss functions: $\ell_{\mathrm{QRE}}$, and $\ell_{\mathrm{quad}}$. All models share the same initialization and the *only* difference here is the loss function. Refer to Appendix H.1 for additional details on the mMLP initialization [6]. The performance is evaluated on the test set, $n_{\mathrm{test}} = 10^3$, in terms of the losses as the error measures, shown as $E_{\mathrm{QRE}}$ and $E_{\mathrm{quad}}$. Table 1 summarizes the results of the evaluation (also refer to Figure 1-B,C for the visual illustration). The key observation is that the quality of estimates differs considerably depending on the choice of the loss function. The loss function $\ell_{\mathrm{QRE}}$ that takes into account the geometry of the SPD matrices clearly outperforms the one based on the Euclidean geometry, $\ell_{\mathrm{qaud}}$. The advantage is more pronounced for small set of training data.

#### 5.1.2 THE MMLP VS. THE MLP BASED ON CHOLESKY DECOMPOSITION

As we discussed in Section 1, one can take a heuristic approach and tailor the output layer of a vanilla MLP so that the resulting output matrix satisfies the SPD requirement.

For the sake of comparison, we solve the same task as in Example 1 using the standard MLP with 3 layers (200 units per layer) and with the quadratic loss. To meet the SPD requirement, we use the

---

[6]Here and in general throughout this section, special care has been made to minimize the effect of overfitting through trying out various initializations and using early stopping.

Table 1: SPD matrix learning on synthetic dataset (Example 1).

| Model/Loss | $d_0=10, n_{\text{train}}=10^4$ | | $d_0=20, n_{\text{train}}=10^4$ | | $d_0=10, n_{\text{train}}=10^2$ | | $d_0=20, n_{\text{train}}=10^2$ | |
|---|---|---|---|---|---|---|---|---|
| | $E_{\text{quad}}$ | $E_{\text{QRE}}$ | $E_{\text{quad}}$ | $E_{\text{QRE}}$ | $E_{\text{quad}}$ | $E_{\text{QRE}}$ | $E_{\text{quad}}$ | $E_{\text{QRE}}$ |
| mMLP/$\ell_{\text{QRE}}$ | $1.2\times10^{-9}$ | $5.8\times10^{-6}$ | $3.3\times10^{-8}$ | $7.7\times10^{-5}$ | $6.3\times10^{-6}$ | $1.1\times10^{-4}$ | $4.3\times10^{-5}$ | $2.2\times10^{-3}$ |
| mMLP/$\ell_{\text{quad}}$ | $2.3\times10^{-4}$ | $2.1\times10^{-2}$ | $2.1\times10^{-3}$ | $3.9\times10^{-2}$ | $3.7\times10^{-2}$ | $6.6\times10^{-1}$ | $6.0\times10^{-2}$ | $1.2$ |
| MLP/$\ell_{\text{quad}}$ | $2.5\times10^{-3}$ | $3.4\times10^{-1}$ | $1.3\times10^{-2}$ | $8.1\times10^{-1}$ | $6.5\times10^{-1}$ | $4.5$ | $7.3\times10^{-1}$ | $6.4$ |

Table 2: Shallow vs deep architecture, Example 2. Error is measured in terms of $E_{\text{QRE}}$.

| Model | $d_0=10$ | | | $d_0=20$ | | |
|---|---|---|---|---|---|---|
| | $L=2$ | $L=4$ | $L=6$ | $L=2$ | $L=4$ | $L=6$ |
| Deep architecture (3) | $2\times10^{-4}$ | $1\times10^{-5}$ | $4\times10^{-5}$ | $5\times10^{-3}$ | $3\times10^{-4}$ | $6\times10^{-4}$ |
| Shallow architecture (18) | $4\times10^{-3}$ | $5\times10^{-3}$ | $4\times10^{-2}$ | $8\times10^{-2}$ | $6\times10^{-2}$ | $7\times10^{-1}$ |

Cholesky decomposition at the output layer using the known result that for a SPD matrix there is a unique lower triangular matrix, with ones as its diagonal entires, and a unique diagonal matrix with positive diagonal entires. Table 1 summarizes the results of the evaluation. Overall the performance is quite poor for small set of training data. As the size of training data grows, the performance improves as expected (refer to Figure 1-D for the visual illustration).

### 5.1.3 SHALLOW VS DEEP SPD MATRIX LEARNING

The design of the mMLP model in (3) enables a mechanism for deep SPD matrix learning by satisfying the SPD constraint across all input, hidden and output layers. The simpler approach would be to consider the standard MLP architecture across input and hidden layers but make use of the activation matrix functions only at the output layer to meet the SPD requirement:

$$\begin{cases} \widehat{\mathbf{Y}} = \mathcal{H}(\mathbf{Z}_0), & \mathbf{Z}_0 = \mathbf{W}_0\mathbf{h}_1(\mathbf{W}_0\mathbf{1})^\top + \mathbf{B}_0, \\ \mathbf{h}_l = \mathfrak{h}(\mathbf{z}_l), & \mathbf{z}_l = \mathbf{W}_l\mathbf{h}_{l+1} + \mathbf{b}_l, \quad 1 \le l \le L, \\ \mathbf{h}_{L+1} = \mathfrak{h}(\mathbf{z}_{L+1}), & \mathbf{z}_{L+1} = \mathbf{W}_{L+1}\text{vec}\mathbf{X} + \mathbf{b}_{L+1}. \end{cases} \quad (18)$$

This amounts to a *shallow design* in the sense that it does not enable a mechanism for preserving the SPD constraint across all layers during the learning.

The design in (3) allows nonlinearities to pass through layers via activation function matrices imposing the SPD constraint, whereas in the shallow design, nonlinearities are propagated across layers via activation functions without imposing any constraints. Our hypothesis is that the former has advantage over the latter in that it captures complex dependencies which are important for the SPD matrix learning. Below, we present a numerical example which indeed highlights the importance of preserving the SPD constraint across all layers when learning the SPD matrix.

**Example 2.** Consider a similar experiment as in Example 1 for the case of $n_{\text{train}} = 10^2$ and output dimensions $d_0 = \{10, 20\}$. We directly compare the performance of (3) against (18) under different number of hidden layers $L = \{2, 4, 6\}$. For the mMLP model, the number of hidden units at each layer is set to 20, and for the MLP model it is set to 200 units. The shallow design (18) uses the hyperbolic tangent as the activation function $\mathfrak{h}(\cdot)$. The same choice of the activation matrix function $\mathcal{H}(\cdot)$, given by (2), is used for both models. We use $\ell_{\text{QRE}}$ as the choice of the loss function for both models (refer to Appendix H.2 for additional details on the initialization). The performance is evaluated in terms of $E_{\text{QRE}}$.

Table 2 summarizes the results of the evaluation. Although the shallow design (18) performs relatively well, it underperforms in comparison to (3). Given the limited number of training samples, arbitrarily increasing the number of layers may not be necessarily advantageous, which is the case for both models. However, in this regard, the design in (18) is more sensitive.

### 5.2 EXPERIMENTAL RESULTS ON THE MULTI-OUTPUT REGRESSION DATASETS

In heteroscedastic multivariate regression, the central task is to estimate the conditional covariance matrix that captures the spatial variations in correlations between the elements of the response vector. The underlying hypothesis is that if a model can effectively capture the heteroscedasticity, then it will provide better uncertainty quantification, and the estimation of the conditional mean response should also improve, in comparison to the model that is build based on the homoscedasticity assumption.

Table 3: Summary of the regression methods used in the experiments.

| | Method | |
|---|---|---|
| Acronym | Source | Model specifications |
| mMLP-mG | Section 4 | 3 layers ($r_l = 200$, $d_l = 20$), Adam optimizer, activation functions as in Table 4 |
| mMLP-mPE | Section 4 | 3 layers ($r_l = 200$, $d_l = 20$), Adam optimizer, activation functions as in Table 4 |
| MLP | Goodfellow et al. (2016) | 3 layers (200 units), ReLU activation, Adam optimizer, and default initialization of the sckit-learn[1] |
| GP | Rasmussen & Williams (2006) | RBF kernel and default initialization of the sckit-learn[1] |
| NBCR | Fox & Dunson (2015) | $10^5$ Gibbs iterations (discarded the first half), truncation of 5, and default initialization of the toolbox[2]. |

[1] **scikit-learn:** https://scikit-learn.org/
[2] **NBCR:** https://homes.cs.washington.edu/ ebfox/software/

Table 4: Choice of activation functions for the mMLP-mPE and mMLP-mG models in Table 3.

| | Output layer | | | | | Hidden layers | |
|---|---|---|---|---|---|---|---|
| Model | $\boldsymbol{\mu}$ | $\log \eta$ | $\alpha$ | $\beta$ | $\Omega$ | $\mathfrak{h}$ | $\mathcal{H}$ |
| mMLP-mG | linear() | linear() | – | – | * | linear() | * |
| mMLP-mPE | linear() | linear() | 0.5+Sigmoid() | 0.5+Sigmoid() | * | linear() | * |

\* **Trace-one Mercer Sigmoid**: $\mathcal{H}(\mathbf{Z}) = \frac{\mathcal{K}(\mathbf{Z})}{\text{tr}(\mathcal{K}(\mathbf{Z}))}$, where $[\mathcal{K}(\mathbf{Z})]_{i,j} = \kappa(\mathbf{z}_i, \mathbf{z}_j)$, $\forall i, j$. The kernel function $\kappa(\cdot, \cdot)$ is given by (2).

Table 5: Datasets and regression results.

(a) Dataset specifications.

| | Dataset | | | | |
|---|---|---|---|---|---|
| Name | Appendix | $d_\mathbf{x}$ | $d_\mathbf{y}$ | $n_{\text{train}}$ | $n_{\text{test}}$ |
| oes10 | H.3.1 | 298 | 16 | 50 | 352 |
| edm | H.3.2 | 2 | 16 | 50 | 103 |
| atp1d | H.3.3 | 411 | 6 | 100 | 236 |
| atp7d | H.3.3 | 411 | 6 | 33 | 303 |
| scm1d | H.3.4 | 280 | 16 | 1000 | 8802 |
| scm20d | H.3.4 | 61 | 16 | 1000 | 8802 |

(b) Prediction errors in terms of the average RMSE across 10 runs. The bold-faced numbers indicate the statistical significance (**p**-value$\leq 0.05$).

| Dataset | mMLP-mPE | mMLP-mG | MLP | GP | NBCR |
|---|---|---|---|---|---|
| oes10 | **0.51** | 0.52 | 0.59 | 0.97 | 1.2 |
| edm | **0.19** | **0.18** | 0.29 | 0.32 | 0.41 |
| atp1d | **0.49** | 0.58 | 0.61 | 0.89 | large |
| atp7d | **0.53** | 0.63 | 0.70 | 0.97 | large |
| scm1d | 0.69 | **0.65** | 0.73 | 0.82 | 1.5 |
| scm20d | **0.67** | 0.72 | 0.78 | 0.87 | 1.7 |

For this purpose, we compare our mMLP-based regression models against another heteroscedasti-based mean-covariance regression model by Fox & Dunson (2015) and two homoscedastic-based mean-regression models, namely, the MLP regressor and the Gaussian process (GP). The models used in this experiment are summarized in Table 3.

**Real-world datasets.** We compare the performance of the heteroscedastic and homoscedastic regression models on six real-world multi-output datasets. Key features of the datasets are summarized in Table 5a. The performance is evaluated in terms of the root-mean-square error (RMSE) on test sets, shown in Table 5b. The result suggests that the mMLP homoscedastic-based regression models are capable of capturing dependencies between the output measurements which contributes to the better estimation of the mean predictions.

## 6    LIMITATIONS AND FUTURE WORK

The main limitation of the mMLP has to do with scalability to higher dimensions. The complexity associated with computing the $\alpha$-derivative of the von Neumann loss function (5) at the output layer is $\mathcal{O}(d_0^3)$. Taking the symmetric nature of the SPD matrices into account, the computational complexity at the hidden layer $l$ reduces to $\mathcal{O}(d_l^2)$.

Our implementation of the matrix backpropagation involves the use of multiple Kronecker products. Although it facilitates the implementation, we would need access to the full Jacobian matrices ($d_l^2 \times d_l^2$). However, these matrices are in fact available in the form of sparse block matrices, which means that it is possible to implement a memory-efficient computation of the tensor products without the need to actually have access to the full matrices. Future work is needed in this direction.

We believe that there are many cases in which the mMLP can prove to be useful. An interesting direction for future work is to investigate application of the model in the context of the conditional density estimation within the framework of variational autoencoders.

## 7    DISCUSSION

We introduced a new method to learn SPD matrices, referred to as the matrix multilayer perceptron (mMLP). The mMLP takes the non-Euclidean geometry of the underlying SPD manifolds into account by making use of the von Neumann divergence as the choice of the SPD manifold metric. One key aspect of the mMLP is that it preserves the SPD constraint across all layers by exploiting PD kernel functions and a backpropagation algorithm that respects the inherent SPD nature of the matrices. We studied application of the model in the context of heteroscedastic multivariate regression. We showed the effectiveness of the proposed model on multiple real-world datasets.

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

## A    MATRIX NOTATIONS

We use $\top$ for the transpose operator, $\text{tr}(\cdot)$ for the trace operator, and $\det(\cdot)$ for the matrix determinant. The symmetric part of a square matrix $\mathbf{B}$ is denoted by $\text{sym}(\mathbf{B}) = (\mathbf{B} + \mathbf{B}^\top)/2$. The Kronecker product is denoted by $\otimes$, the Hadamard product by $\circ$, and the dot product by $\odot$. We use the vec-operator for *column-by-column* stacking of a matrix $\mathbf{A}$, shown as $\text{vec}\mathbf{A} \equiv \text{vec}(\mathbf{A})$. Let $\mathbf{A}$ be an $m \times n$ matrix, the operator $\mathsf{P}_{(m,n)}$ will then rearrange $\text{vec}\mathbf{A}$ to its matrix form: $\mathbf{A} = \mathsf{P}_{(m,n)}(\text{vec}\mathbf{A})$. For the $m \times n$ dimensional matrix $\mathbf{A}$, the commutation matrix is shown as $\boldsymbol{K}_{(m,n)}$ which is the $mn \times mn$ matrix that transforms $\text{vec}\mathbf{A}$ into $\text{vec}\mathbf{A}^\top$ as: $\boldsymbol{K}_{(m,n)}\text{vec}\mathbf{A} = \text{vec}\mathbf{A}^\top$. An $m \times m$ identity matrix is shown as $\boldsymbol{I}_m$. If $\mathcal{T}(\mathbf{X}) : \mathbb{R}^{d \times d} \to \mathbb{R}$ is a real-valued function on matrices, then $\nabla_\mathbf{X}\mathcal{T}(\mathbf{X})$ denotes the gradient with respect to the matrix $\mathbf{X}$, $\nabla_\mathbf{X}\mathcal{T}(\mathbf{X}) = \left[\frac{\partial \mathcal{T}}{\partial \mathbf{x}_{ij}}\right]_{i,j=1:d}$. The *matrix logarithm* and the *matrix exponential* are written as $\mathfrak{log}\mathbf{A}$ and $\mathfrak{exp}\mathbf{A}$, respectively. The matrix exponential in the case of symmetric matrices can be expressed using the eigenvalue decomposition as $\mathfrak{exp}\mathbf{A} = \mathbf{V}(\mathfrak{exp}\mathbf{\Lambda})\mathbf{V}^\top$, where $\mathbf{V}$ is an orthonormal matrix of eigenvectors and $\mathbf{\Lambda}$ is a diagonal matrix with the eigenvalues on the diagonal. The matrix logarithm is the inverse of the matrix exponential if it exists. If $\mathbf{A}$ is symmetric and strictly positive definite (PD) it is computed using $\mathfrak{log}\mathbf{A} = \mathbf{V}(\log\mathbf{\Lambda})\mathbf{V}^\top$, where $(\log\mathbf{\Lambda})_{i,i} = \log\Lambda_{i,i}$.

## B    THE $\alpha$-DERIVATIVE: DEFINITION AND PROPERTIES

**Definition**    Let $\mathbf{F}$ be an $m \times n$ matrix function of an $n \times q$ matrix of variables $\mathbf{X}$. The $\alpha$-derivative of $\mathbf{F}(\mathbf{X})$ is defined as (Magnus, 2010, Definition 2)

$$\mathsf{D}_\mathbf{X}\mathbf{F} := \frac{\partial\,\text{vec}\mathbf{F}(\mathbf{X})}{\partial\,(\text{vec}\mathbf{X})^\top}, \tag{19}$$

where $\mathsf{D}_\mathbf{X}\mathbf{F}$ is an $mp \times nq$ matrix which contains all the partial derivatives such that each row contains the partial derivatives of one function with respect to all variables, and each column contains the partial derivatives of all functions with respect to one variable.

**Product rule**    Let $\mathbf{F}$ $(m \times p)$ and $\mathbf{G}$ $(p \times r)$ be functions of $\mathbf{X}$ $(n \times q)$. Then the product rule for the $\alpha$-derivative is given by (Magnus, 2010)

$$\mathsf{D}_\mathbf{X}(\mathbf{FG}) = (\mathbf{G}^\top \otimes \boldsymbol{I}_m)\mathsf{D}_\mathbf{X}\mathbf{F} + (\boldsymbol{I}_r \otimes \mathbf{F})\mathsf{D}_\mathbf{X}\mathbf{G}. \tag{20}$$

**Chain rule**    Let $\mathbf{F}$ $(m \times p)$ be differentiable at $\mathbf{X}$ $(n \times q)$, and $\mathbf{G}$ $(l \times r)$ be differentiable at $\mathbf{Y} = \mathbf{F}(\mathbf{X})$, then the composite function $\mathbf{H}(\mathbf{X}) = \mathbf{G}(\mathbf{F}(\mathbf{X}))$ is differentiable at $\mathbf{X}$, and

$$\mathsf{D}_\mathbf{X}\mathbf{H} = \mathsf{D}_\mathbf{Y}\mathbf{G}\mathsf{D}_\mathbf{X}\mathbf{F}, \tag{21}$$

which expresses the chain rule for the $\alpha$-derivative (Magnus, 2010).

## C    THE BASIC CASE OF THE MMLP

### C.1    FEEDFORWARD STEP

At the feed-forward computation, we compute and store the latent outputs $\widehat{\mathbf{Y}}$, $\mathbf{H}_l$ for all $l \in \{L + 1, \ldots, 1\}$ using the current setting of the parameters, which are $\mathbf{W}_l$, and $\mathbf{B}_l$ computed from the learning step, Appendix C.3.

### C.2    BACKPROPAGATION STEP

We first summarize the necessary $\alpha$-derivatives for the backpropagation, and then write down the backpropagation procedure accordingly.

**Derivatives required for backpropagation**    The derivative of the activation matrix function depends on the specific choice of kernel function, and in general it is computed readily from the

definition of $\alpha$-derivative,

$$\mathsf{D}_{\mathbf{Z}_l} \mathcal{H}(\mathbf{Z}_l) := \frac{\partial \, \mathrm{vec} \mathcal{H}(\mathbf{Z}_l)}{\partial \, (\mathrm{vec} \mathbf{Z}_l)^\top}, \qquad l \in \{0, \dots, L+1\}. \tag{22}$$

For our specific choice of activation function, the Mercer Sigmoid kernel (2), it is computed in Appendix E.

Via repeated use of the product rule of $\alpha$-derivatives (20), we obtain

$$\mathsf{D}_{\mathbf{W}_l} \mathbf{Z}_l = (\mathbf{W}_l \otimes \boldsymbol{I}_{d_l})(\mathbf{H}_{l+1}^\top \otimes \boldsymbol{I}_{d_l}) + (\boldsymbol{I}_{d_l} \otimes (\mathbf{W}_l \mathbf{H}_{l+1})) \boldsymbol{K}_{(d_l, d_{l+1})}, \quad l \in \{0, \dots, L\}, \tag{23}$$

$$\mathsf{D}_{\mathbf{W}_{L+1}} \mathbf{Z}_l = (\mathbf{W}_{L+1} \mathbf{1}_{p_1 p_2} \otimes \boldsymbol{I}_{d_{L+1}})((\mathrm{vec} \mathbf{X})^\top \otimes \boldsymbol{I}_{d_{L+1}})$$
$$+ (\boldsymbol{I}_{d_{L+1}} \otimes (\mathbf{W}_{L+1} \mathrm{vec} \mathbf{X}))(\boldsymbol{I}_{d_{L+1}} \otimes \mathbf{1}_{p_1 p_2}^\top) \boldsymbol{K}_{(d_{L+1}, p_1 p_2)}, \tag{24}$$

$$\mathsf{D}_{\mathbf{H}_{l+1}} \mathbf{Z}_l = (\mathbf{W}_l \otimes \boldsymbol{I}_{d_l})(\boldsymbol{I}_{d_{l+1}} \otimes \mathbf{W}_l), \quad l \in \{0, \dots, L\}, \tag{25}$$

where $\boldsymbol{K}$ is the commutation matrix (refer to Appendix A for a summary of the matrix notation).

**Backpropagation**   In the interest of simple expressions, let $\mathbf{H}_0 \equiv \widehat{\mathbf{Y}}$. Backpropagation to the hidden layer $l$ is computed recursively using the derivatives computed at the previous layer according to

$$\mathsf{D}_{\mathbf{Z}_l} \ell = \mathsf{D}_{\mathbf{H}_l} \ell \mathsf{D}_{\mathbf{Z}_l} \mathbf{H}_l, \qquad\qquad \forall l \in \{0, \dots, L+1\}, \tag{26}$$

$$\mathsf{D}_{\mathbf{W}_l} \ell = \mathsf{D}_{\mathbf{Z}_l} \ell \mathsf{D}_{\mathbf{W}_l} \mathbf{Z}_l, \qquad\qquad \forall l \in \{0, \dots, L+1\}, \tag{27}$$

$$\mathsf{D}_{\mathbf{H}_{l+1}} \ell = \mathsf{D}_{\mathbf{Z}_l} \ell \mathsf{D}_{\mathbf{H}_{l+1}} \mathbf{Z}_l, \qquad\qquad \forall l \in \{0, \dots, L\}, \tag{28}$$

$$\mathsf{D}_{\mathbf{B}_l} \ell = \mathsf{D}_{\mathbf{Z}_l} \ell, \qquad\qquad \forall l \in \{0, \dots, L+1\}. \tag{29}$$

## C.3   LEARNING STEP

Learning involves updating the weights $\mathbf{W}_l$ and the biases $\mathbf{B}_l$ using derivatives computed during the backpropagation step. These are updated using derivatives $\mathsf{D}_{\mathbf{W}_l} \ell$ and $\mathsf{D}_{\mathbf{B}_l} \ell$ for a given learning rate $\eta$ as

$$\mathbf{W}_l \leftarrow \mathbf{W}_l - \eta \mathsf{P}_{(d_l, d_{l+1})}(\mathsf{D}_{\mathbf{W}_l} \ell), \qquad\qquad \forall l \in \{0, \dots, L+1\}, \tag{30}$$

$$\mathbf{B}_l \leftarrow \mathbf{B}_l - \eta \mathsf{P}_{(d_l, d_l)}(\mathsf{D}_{\mathbf{B}_l} \ell), \qquad\qquad \forall l \in \{0, \dots, L+1\}, \tag{31}$$

where $\mathsf{P}$ is the rearrangement operator introduced in Appendix A.

## D   THE GENERAL FORM OF THE MMLP

### D.1   FEEDFORWARD STEP

The forward path involves computing and storing both $\mathbf{h}_l, \widehat{\mathbf{y}}$ and $\mathbf{H}_l, \widehat{\mathbf{Y}}$ using the current settings of the parameters, for all $l \in \{0, \dots, L+1\}$.

### D.2   BACKPROPAGATION STEP

Most of the necessary derivatives are identical to the ones in Appendix C.2. However, there are some additional derivatives needed which we will discuss in the following. We then write down the backpropagation formula.

### D.2.1   REQUIRED DERIVATIVES FOR BACKPROPAGATION

The derivative of the activation function depends on the choice of the function, and it is computed using the definition of the $\alpha$-derivative,

$$\mathsf{D}_{\mathbf{z}_l} \mathfrak{h}_l(\mathbf{z}_l) = \frac{\partial \, \mathfrak{h}(\mathbf{z}_l)}{\partial \, (\mathbf{z}_l)^\top}, \quad l \in \{0, \dots, L+1\}. \tag{32}$$

The other required derivatives are computed as

$$D_{\mathbf{A}_l}\mathbf{z}_l = \mathbf{C}_l\mathbf{H}_l, \qquad\qquad l \in \{0,\ldots,L\}, \qquad (33)$$

$$D_{\mathbf{A}_{L+1}}\mathbf{z}_{L+1} = (\mathbf{C}_{L+1}\mathbf{H}_{L+1})(\mathbf{1}_{r_{L+1}}^\top \otimes \boldsymbol{I}_{d_{L+1}}), \qquad (34)$$

$$D_{\mathbf{C}_l}\mathbf{z}_l = (\mathbf{H}_l\mathbf{A}_l\mathbf{h}_{l+1})^\top \otimes \boldsymbol{I}_{r_l}, \qquad\qquad l \in \{0,\ldots,L\}, \qquad (35)$$

$$D_{\mathbf{C}_{L+1}}\mathbf{z}_{L+1} = (\mathbf{H}_{L+1}\mathbf{A}_{L+1}\mathbf{1}_{r_{L+1}})^\top \otimes \boldsymbol{I}_{r_{L+1}}, \qquad (36)$$

$$D_{\mathbf{h}_{l+1}}\mathbf{z}_l = \mathbf{C}_l\mathbf{H}_l\mathbf{A}_l, \qquad\qquad l \in \{0,\ldots,L\}, \qquad (37)$$

$$D_{\mathbf{H}_l}\mathbf{z}_l = ((\mathbf{A}_l\mathbf{h}_{l+1})^\top \otimes \boldsymbol{I}_{r_l})(\boldsymbol{I}_{d_l} \otimes \mathbf{C}_l), \qquad\qquad l \in \{0,\ldots,L\}, \qquad (38)$$

$$D_{\mathbf{H}_{L+1}}\mathbf{z}_{L+1} = ((\mathbf{A}_{L+1}\mathbf{1}_{L+1})^\top \otimes \boldsymbol{I}_{r_{L+1}})(\boldsymbol{I}_{d_{L+1}} \otimes \mathbf{C}_{L+1}). \qquad (39)$$

**Backpropagation** For simplicity of expressions, let $\mathbf{h}_0 \equiv \widehat{\mathbf{y}}$ and $\mathbf{H}_0 \equiv \widehat{\mathbf{Y}}$. The derivatives are recursively computed as

$$D_{\mathbf{h}_0}\ell \equiv D_{\widehat{\mathbf{y}}}\ell \qquad (40)$$

$$D_{\mathbf{H}_0}\ell \equiv D_{\widehat{\mathbf{Y}}}\ell = D_{\mathbf{z}_0}\ell D_{\widehat{\mathbf{Y}}}\mathbf{z}_0 + D_{\widehat{\mathbf{y}}}\ell \qquad (41)$$

$$D_{\mathbf{z}_l}\ell = D_{\mathbf{h}_l}\ell D_{\mathbf{z}_l}\mathbf{h}_l, \qquad\qquad \forall l \in \{0,\ldots,L+1\}, \qquad (42)$$

$$D_{\mathbf{h}_{l+1}}\ell = D_{\mathbf{z}_l}\ell D_{\mathbf{h}_{l+1}}\mathbf{z}_l, \qquad\qquad \forall l \in \{0,\ldots,L\}, \qquad (43)$$

$$D_{\mathbf{Z}_l}\ell = D_{\mathbf{H}_l}\ell D_{\mathbf{Z}_l}\mathbf{H}_l, \qquad\qquad \forall l \in \{0,\ldots,L+1\}, \qquad (44)$$

$$D_{\mathbf{H}_{l+1}}\ell = D_{\mathbf{z}_{l+1}}\ell D_{\mathbf{H}_{l+1}}\mathbf{z}_{l+1} + D_{\mathbf{z}_l}\ell D_{\mathbf{H}_{l+1}}\mathbf{Z}_l, \qquad\qquad \forall l \in \{0,\ldots,L\}, \qquad (45)$$

$$D_{\mathbf{A}_l}\ell = D_{\mathbf{z}_l}\ell D_{\mathbf{A}_l}\mathbf{z}_l, \qquad\qquad \forall l \in \{0,\ldots,L+1\}, \qquad (46)$$

$$D_{\mathbf{C}_l}\ell = D_{\mathbf{z}_l}\ell D_{\mathbf{C}_l}\mathbf{z}_l, \qquad\qquad \forall l \in \{0,\ldots,L+1\}, \qquad (47)$$

$$D_{\mathbf{W}_l}\ell = D_{\mathbf{z}_l}\ell D_{\mathbf{W}_l}\mathbf{Z}_l, \qquad\qquad \forall l \in \{0,\ldots,L+1\}, \qquad (48)$$

$$D_{\mathbf{b}_l}\ell = D_{\mathbf{z}_l}\ell, \qquad\qquad \forall l \in \{0,\ldots,L+1\}, \qquad (49)$$

$$D_{\mathbf{B}_l}\ell = D_{\mathbf{Z}_l}\ell, \qquad\qquad \forall l \in \{0,\ldots,L+1\}. \qquad (50)$$

### D.3 LEARNING STEP

The learning step involves updating the weights and the biases which are computed using derivatives computed from the backpropagation step. Update rules for $\mathbf{W}_l$, and $\mathbf{B}_l$ are the same as the ones given in Appendix C.3. The remaining parameters are learned in a similar fashion,

$$\mathbf{A}_l \leftarrow \mathbf{A}_l - \eta \mathsf{P}_{(d_l,r_{l+1})}(D_{\mathbf{A}_l}\ell), \qquad\qquad \forall l \in \{0,\ldots,L+1\}, \qquad (51)$$

$$\mathbf{C}_l \leftarrow \mathbf{C}_l - \eta \mathsf{P}_{(r_l,d_l)}(D_{\mathbf{C}_l}\ell), \qquad\qquad \forall l \in \{0,\ldots,L+1\}, \qquad (52)$$

$$\mathbf{b}_l \leftarrow \mathbf{b}_l - \eta \mathsf{P}_{(r_l,1)}(D_{\mathbf{b}_l}\ell), \qquad\qquad \forall l \in \{0,\ldots,L+1\}. \qquad (53)$$

## E THE $\alpha$-DERIVATIVE OF THE MERCER SIGMOID KERNEL

The $\alpha$-derivative of the Mercer sigmoid kernel can be computed as

$$D_{\mathbf{Z}}\mathcal{H} = \begin{pmatrix} \cdots & \cdots & \cdots \\ \cdots & \underbrace{\dfrac{\partial}{\partial \mathbf{z}_i} \dfrac{\kappa_{mn}}{\mathrm{tr}\mathcal{K}}}_{1 \times d_l} & \cdots \\ \cdots & \cdots & \cdots \end{pmatrix}, \qquad \forall\, i, m, n \in \{1,\ldots,d_l\}, \qquad (54)$$

where $\mathbf{z}_i$ indicates the $i^{\text{th}}$ column of $\mathbf{Z}$, $\mathrm{tr}\mathcal{K} \equiv \mathrm{tr}\mathcal{K}(\mathbf{Z})$, $\kappa_{mn} \equiv \kappa(\mathbf{z}_m, \mathbf{z}_n)$ as defined in (2), and

$$\frac{\partial}{\partial \mathbf{z}_i} \frac{\kappa_{mn}}{\mathrm{tr}\mathcal{K}} = \left(\frac{\alpha(\mathbf{1}^\top - \mathfrak{f}(\mathbf{z}_i) \circ \mathfrak{f}(\mathbf{z}_i))}{(\mathrm{tr}\mathcal{K})^2}\right) \circ \left(\mathrm{tr}\mathcal{K}\,\mathfrak{f}\left(\frac{\partial}{\partial \mathbf{z}_i}(\mathbf{z}_m \circ \mathbf{z}_n)\right) - 2\kappa_{mn}\mathfrak{f}(\mathbf{z}_i)\right), \qquad (55)$$

where $\mathfrak{f}(\mathbf{z}_i) := \tanh(\alpha \mathbf{z}_i - \beta)$.

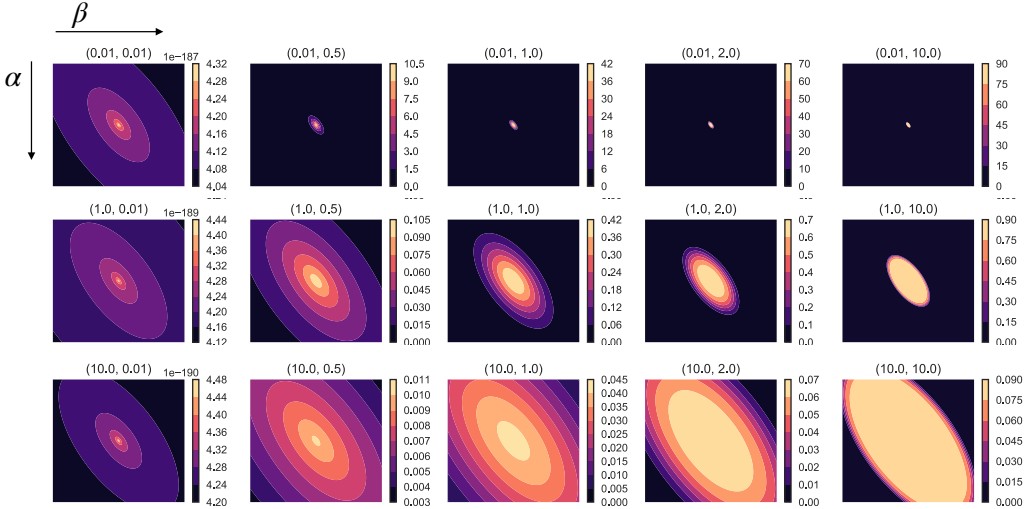

Figure G.1: The probability density function of a trace-one mPE distribution, $\mathcal{E}_{\text{tr1}}(\boldsymbol{\mu}, \boldsymbol{M}/\eta, \eta, \alpha, \beta)$ for fixed values of $\boldsymbol{\mu}, \boldsymbol{M}, \eta = \text{tr}(\boldsymbol{M})$ and varying values of scale and shape parameters $(\alpha, \beta)$. When $\alpha = 1$ and $\beta = 1$, the density corresponds to the multivariate Gaussian distribution $\mathcal{N}(\boldsymbol{\mu}, \boldsymbol{M})$.

## F    TRACE-ONE MULTIVARIATE GAUSSIAN DISTRIBUTION

### F.1    PROBABILITY DENSITY FUNCTION

For a $d$-dimensional random variable $\boldsymbol{\vartheta} \in \mathbb{R}^d$, we define the trace-one Gaussian distribution according to

$$\mathcal{N}_{\text{tr1}}(\boldsymbol{\vartheta}; \boldsymbol{\mu}, \boldsymbol{\Omega}, \eta) = \frac{1}{\det(2\pi\eta\boldsymbol{\Omega})^{\frac{1}{2}}} e^{-\frac{1}{2}(\boldsymbol{\vartheta}-\boldsymbol{\mu})^{\top}(\eta\boldsymbol{\Omega})^{-1}(\boldsymbol{\vartheta}-\boldsymbol{\mu})}, \tag{56}$$

where $\boldsymbol{\mu} \in \mathbb{R}^d$ is the mean, $\eta \in \mathbb{R}^+$ is the scale parameter, and $\boldsymbol{\Omega} \in \mathbb{R}^{d \times d}$ is the trace-one covariance matrix, $\text{tr}(\boldsymbol{\Omega}) = 1$.

### F.2    THE $\alpha$-DERIVATIVES OF THE LOG-PDF

The $\alpha$-derivatives of the trace-one Gaussian distribution's log-pdf with respect to its parameters are summarized as

$$D_{\boldsymbol{\Omega}}\log\mathcal{N}_{\text{tr1}}(\boldsymbol{\vartheta}; \boldsymbol{\mu}, \boldsymbol{\Omega}, \eta) = -\frac{1}{2}(\text{vec}(\boldsymbol{\Omega}^{-1}))^{\top} - \frac{1}{2}(D_{\boldsymbol{\Omega}}t)^{\top}, \tag{57}$$

$$D_{\boldsymbol{\Omega}}t = -\text{vec}((\eta\boldsymbol{\Omega})^{-1}(\boldsymbol{\vartheta} - \boldsymbol{\mu})(\boldsymbol{\vartheta} - \boldsymbol{\mu})^{\top}\boldsymbol{\Omega}^{-1}), \tag{58}$$

$$D_{\boldsymbol{\mu}}\log\mathcal{N}_{\text{tr1}}(\boldsymbol{\vartheta}; \boldsymbol{\mu}, \boldsymbol{\Omega}, \eta) = (\boldsymbol{\vartheta} - \boldsymbol{\mu})(\eta\boldsymbol{\Omega})^{-1}, \tag{59}$$

$$D_{\log\eta}\log\mathcal{N}_{\text{tr1}}(\boldsymbol{\vartheta}; \boldsymbol{\mu}, \boldsymbol{\Omega}, \eta) = -\frac{d}{2} + \frac{1}{2}(\boldsymbol{\vartheta} - \boldsymbol{\mu})^{\top}(\eta\boldsymbol{\Omega})^{-1}(\boldsymbol{\vartheta} - \boldsymbol{\mu}). \tag{60}$$

## G    TRACE-ONE MULTIVARIATE POWER EXPONENTIAL (MPE) DISTRIBUTION

### G.1    PROBABILITY DENSITY FUNCTION

For a random variable $\boldsymbol{\vartheta} \in \mathbb{R}^d$, the trace-one mPE distribution can be expressed as

$$\mathcal{E}_{\text{tr1}}(\boldsymbol{\vartheta}; \boldsymbol{\mu}, \boldsymbol{\Omega}, \alpha, \beta, \eta) = \frac{c(\alpha, \beta)}{(\det(\eta\boldsymbol{\Omega}))^{\frac{1}{2}}} \exp\left\{-\frac{1}{2}\left(\frac{t(\boldsymbol{\vartheta}; \boldsymbol{\mu}, \boldsymbol{\Omega})}{\alpha\eta}\right)^{\beta}\right\},$$

$$c(\alpha, \beta) = \frac{\beta\Gamma(\frac{d}{2})}{\pi^{\frac{d}{2}}\Gamma(\frac{d}{2\beta})2^{\frac{d}{2\beta}}\alpha^{\frac{d}{2}}}, \quad t(\boldsymbol{\vartheta}; \boldsymbol{\mu}, \boldsymbol{\Omega}) := (\boldsymbol{\vartheta} - \boldsymbol{\mu})^{\top}\boldsymbol{\Omega}^{-1}(\boldsymbol{\vartheta} - \boldsymbol{\mu}), \quad \text{tr}(\boldsymbol{\Omega}) = 1. \tag{61}$$

Here, $\boldsymbol{\mu}$ is the mean vector, and $\boldsymbol{\Omega}$ is a $d \times d$ symmetric real dispersion matrix where $\text{tr}(\boldsymbol{\Omega}) = 1$. The parameter $\eta$ has the same role as in the trace-one Gaussian distribution. The pair of $\alpha \in \mathbb{R}^+$

and $\beta \in \mathbb{R}^+$ control the tail and the shape of the distribution. As a special case, the mPE includes the Gaussian distribution: For $\alpha = 1$ and $\beta = 1$, the trace-one mPE distribution corresponds to the trace-one multivariate Gaussian distribution. Figure G.1.1 visualizes the probability density function of the distribution for selected values of $\alpha$ and $\beta$, for the case of $d = 2$.

*Remark:* Very large and very small values of $\alpha$ and $\beta$ might be undesirable as, for one, they pose numerical challenges. In practice, these parameters can be restricted within a range by choosing suitable output activation functions. In all experiments in this paper, we choose to bound them conservatively as: $0.5 \leq \alpha, \beta \leq 1.5$, using the sigmoid function.

### G.2  MOMENTS

Let $\boldsymbol{\vartheta} \in \mathbb{R}^d$ and $\boldsymbol{\vartheta} \sim \mathcal{E}_{\mathrm{tr1}}(\boldsymbol{\mu}, \boldsymbol{\Omega}, \alpha, \beta, \eta)$. The mPE's mean vector and covariance matrix are computed from:

$$\mathbb{E}[\boldsymbol{\vartheta}] = \boldsymbol{\mu}, \qquad \mathbb{V}[\boldsymbol{\vartheta}] = \alpha \eta \nu(\beta) \boldsymbol{\Omega}, \qquad \nu(\beta) := \frac{2^{1/\beta} \Gamma(\frac{d+2}{2\beta})}{d \Gamma(\frac{d}{2\beta})}, \tag{62}$$

where $\Gamma(\cdot)$ denotes the gamma function.

### G.3  THE $\alpha$-DERIVATIVES OF THE LOG-PDF

It is straightforward to take derivatives of the mPE's log-pdf using the favorable generalization properties of the $\alpha$-derivative's chain and product rules. These are summarized as:

$$\mathsf{D}_{\boldsymbol{\Omega}} \log \mathcal{E}_{\mathrm{tr1}}(\boldsymbol{\vartheta}; \boldsymbol{\mu}, \boldsymbol{\Omega}, \alpha, \beta, \eta) = -\frac{1}{2}(\mathrm{vec}(\boldsymbol{\Omega}^{-\top}))^\top - \frac{\beta}{2\alpha\eta}(t/\alpha\eta)^{\beta-1} \mathsf{D}_{\boldsymbol{\Omega}} t, \tag{63}$$

$$\mathsf{D}_{\boldsymbol{\Omega}} t = -(\mathrm{vec}(\boldsymbol{\Omega}^{-1}(\boldsymbol{\vartheta}-\boldsymbol{\mu})(\boldsymbol{\vartheta}-\boldsymbol{\mu})^\top \boldsymbol{\Omega}^{-1})^\top)^\top, \tag{64}$$

$$\mathsf{D}_{\boldsymbol{\mu}} \log \mathcal{E}_{\mathrm{tr1}}(\boldsymbol{\vartheta}; \boldsymbol{\mu}, \boldsymbol{\Omega}, \alpha, \beta, \eta) = -\frac{\beta}{2\alpha\eta}(t/\alpha\eta)^{\beta-1} \mathsf{D}_{\boldsymbol{\mu}} t, \tag{65}$$

$$\mathsf{D}_{\boldsymbol{\mu}} t = -2(\mathrm{vec}((\boldsymbol{\vartheta}-\boldsymbol{\mu})^\top \boldsymbol{\Omega}^{-1})^\top)^\top, \tag{66}$$

$$\mathsf{D}_{\log\eta} \log \mathcal{E}_{\mathrm{tr1}}(\boldsymbol{\vartheta}; \boldsymbol{\mu}, \boldsymbol{\Omega}, \alpha, \beta, \eta) = -\frac{d}{2} + \frac{\beta t}{2\alpha\eta}(t/\alpha\eta)^{\beta-1}, \tag{67}$$

$$\mathsf{D}_{\alpha} \log \mathcal{E}_{\mathrm{tr1}}(\boldsymbol{\vartheta}; \boldsymbol{\mu}, \boldsymbol{\Omega}, \alpha, \beta, \eta) = -\frac{d}{2\alpha} + \frac{\beta t}{2\eta\alpha^2}(t/\alpha\eta)^{\beta-1}, \tag{68}$$

$$\mathsf{D}_{\beta} \log \mathcal{E}_{\mathrm{tr1}}(\boldsymbol{\vartheta}; \boldsymbol{\mu}, \boldsymbol{\Omega}, \alpha, \beta, \eta) = \mathsf{D}_{\beta} \log c(\alpha, \beta) - \frac{1}{2}(t/\alpha\eta)^\beta \log(t/\alpha\eta), \tag{69}$$

$$\mathsf{D}_{\beta} \log c(\alpha, \beta) = \frac{1}{\beta} + \frac{d}{2\beta^2}(\psi(d/2\beta) + \log 2). \tag{70}$$

## H  ADDITIONAL DETAILS ON THE EXPERIMENTS

### H.1  EXAMPLE 1

**Data generation**  Let $\boldsymbol{A} \in \mathbb{R}^{d_0 \times d_0}$ be a matrix where each of its elements is generated from a standard normal distribution. The matrix $\boldsymbol{A}$ is kept fixed. The $i^{\mathrm{th}}$ class covariance $\mathbf{Y}_i$ is computed according to the following procedure:

1. Draw $10^4$ samples from a known Gaussian distribution $\mathcal{N}(\boldsymbol{\mu}_i, \boldsymbol{\Sigma}_i)$ with a unique mean $\boldsymbol{\mu}_i \in \mathbb{R}^{d_0}$ and a unique dense covariance matrix $\boldsymbol{\Sigma}_i \in \mathbb{R}^{d_0 \times d_0}$.
2. Let $\boldsymbol{t}_j$ be a random sample from this Gaussian. For this sample, compute $\boldsymbol{y}_j = \boldsymbol{A}\boldsymbol{t}_j$. For all $10^4$ samples, collect the results into $\underline{\boldsymbol{y}} = \{\boldsymbol{y}_j\}_{j=1}^{10^4}$.
3. Compute the sample covariance of $\underline{\boldsymbol{y}}$ and normalize the resulting covariance matrix to trace one, that is $\mathbf{Y}_i \leftarrow \mathrm{cov}(\underline{\boldsymbol{y}})/\mathrm{tr}(\mathrm{cov}(\underline{\boldsymbol{y}}))$.

**Initialization** All models use the same batch size (equal to 5), the same choice of activation matrix function, which is given by the Mercer sigmoid kernel (2), and the same optimizer (the Adam optimizer (Kingma & Welling, 2014) with default settings).

## H.2 EXAMPLE 2

**Data generation** See the data generation procedure in Example 1.

**Initialization** Both models (18) and (3) use the same batch size (equal to 5), the same choice of loss function (5), and the same optimizer (the Adam optimizer (Kingma & Welling, 2014) with default settings). Both models use the same choice of the output activation matrix function, given by the Mercer sigmoid kernel (2). The model in (18) uses the hyperbolic tangent as the activation function across the hidden layers, while (3) makes use of the same choice of the activation matrix function as in its output layer.

## H.3 MULTI-OUTPUT REGRESSION DATASETS

### H.3.1 oes10

The dataset oes10 was obtained from Spyromitros-Xioufis et al. (2016). The Occupational Employment Survey (OES) datasets contain records from the years of 2010 (OES10) of the annual Occupational Employment Survey compiled by the US Bureau of Labor Statistics. As described in (Spyromitros-Xioufis et al., 2016), "each row provides the estimated number of full-time equivalent employees across many employment types for a specific metropolitan area". We selected the same 16 target variables as listed in (Spyromitros-Xioufis et al., 2016, Table 5). The remaining 298 variables serve as the inputs. Data samples were randomly divided into training and test sets (refer to Table 5a).

### H.3.2 edm

The dataset edm was obtained from Karalic & Bratko (1997). The electrical discharge machining (EDM) dataset contain domain measurements in which the workpiece surface is machined by electrical discharges occurring in the gap between two electrodes: the tool and the workpiece. Given the two input variables, gap control and flow control, the aim here is to predict the other 16 target variables representing mean values and deviations of the observed quantities of the considered machining parameters. Data samples were randomly divided into training and test sets (refer to Table 5a).

### H.3.3 atp1d AND atp7d

The datasets atp1d and atp7d were obtained from (Spyromitros-Xioufis et al., 2016). The Airline Ticket Price (ATP) dataset includes the prediction of airline ticket prices. As described in (Spyromitros-Xioufis et al., 2016), the target variables are either the next day price, atp1d, or minimum price observed over the next seven days atp7d for 6 target flight preferences listed in (Spyromitros-Xioufis et al., 2016, Table 5). There are 411 input variables in each case. The inputs for each sample are values considered to be useful for prediction of the airline ticket prices for a specific departure date, for example, the number of days between the observation date and the departure date, or the boolean variables for day-of-the-week of the observation date. Data samples were randomly divided into training and test sets (refer to Table 5a).

### H.3.4 scm1d AND scm20d

The datasets scm1d and scm20d were obtained from (Spyromitros-Xioufis et al., 2016). The Supply Chain Management (SCM) datasets are derived from the Trading Agent Competition in Supply Chain Management (TAC SCM) tournament from 2010. As described in (Spyromitros-Xioufis et al., 2016), each row corresponds to an observation day in the tournament. There are 280 input variables in these datasets which are observed prices for a specific tournament day. The datasets contain 16 regression targets, where each target corresponds to the next day mean price scm1d or mean price for 20 days in the future scm20d for each product (Spyromitros-Xioufis et al., 2016, Table 5). Data samples were randomly divided into training and test sets (refer to Table 5a).

