# OpenReview forum: "Matrix Multilayer Perceptron"
_ICLR.cc/2020/Conference — Reject_

### Official Review · AnonReviewer2 · 2019-10-22
**Official Blind Review #2**

**Rating:** 6

**Review:**

This paper generalized neural networks into case where a semidefinite positive matrix is learned at the output. The paper presents theoretical derivations that look sound, and validating experiments on synthetic and real data. I must say my expertise does not really correspond to what is done in this paper, but I do not see any obvious flaws and the results look solid. I appreciated the discussion of limitations in section 6. I vote for acceptance with the weakest possible confidence level since it is likely I missed many important points.

**Experience Assessment:**

I do not know much about this area.

**Review Assessment: Checking Correctness Of Derivations And Theory:**

I did not assess the derivations or theory.

**Review Assessment: Checking Correctness Of Experiments:**

I assessed the sensibility of the experiments.

**Review Assessment: Thoroughness In Paper Reading:**

I made a quick assessment of this paper.

---

> ### Author Response · Authors · 2019-11-14
> **Thank you for reviewing our paper.**
>
> Thank you for reviewing our paper.

---

### Official Review · AnonReviewer3 · 2019-10-23
**Official Blind Review #3**

**Rating:** 6

**Review:**

This manuscript proposes a novel formulation of the MLP to address predicting a symmetric positive definite (SPD) matrix from an input vector or matrix.  While the field has had methods for years to estimate SPD matrices (such as the covariance matrix estimate in the reparameterization trick), this manuscript proposes a markedly different approach based on a different layer structure and repeated normalization steps based on Mercer Kernels.  Additionally, the loss used can be modified to use more PSD-specific losses, such as the symmetrized von Neumann divergence, rather than the traditional quadratic loss.  This loss appears to give significantly better solutions on synthetic data.

While there is some interesting and potentially useful novelty in the approach, I have some concerns about the empirical evidence and modeling to truly determine the mMLP's utility.

First, in the synthetic data the mMLP with the l_QRE loss outperforms the l_quad loss with regards to both E_QRE and E_quad.  Why does mMLP/l_QRE outperform on E_quad? As l_quad is actually designed to minimize this error, this is surprising and needs additional explanation.  Additionally, there are two major changes between mMLP and the MLP.  One is the network structure, and the second is the trace being normalized; which change really induces the improvements in the performance? If the l_QRE loss was used in the MLP, would you get similar improvements?

The network structure as a whole needs greater validation.  A major difference in (3) is that the weight matrices W_l are applied as a both a left and right multiplication.  Given that the \mathcal{H} operation symmetrizes and normalizes the matrix, a symmetric operation isn't strictly necessary here.  Using both multiplications leads to quadratic properties, which in my experience are less stable in the optimization.  Can the authors validate this structure versus the simpler structure of simply using left multiplications?  Or, in other words, is this weight multiplication structure helpful or do the benefits really come from the \mathcal{H} operation.

I think that the heteroscedastic regression experiments don't evaluate on one of the key issues, which is uncertainty estimation.  The trace-1 normalization is highly restrictive, so I imagine that that this method is getting the uncertainty incorrect.  Also, heteroscedastic regression has a long history in neural networks, dating back to at least Nix and Wiegand in 1994.  The manuscript needs to be updated to reflect the historical work and current literature on the topic.

Please check Table 5(a), which states that you are only using a small number of training samples.  Also, given the relatively small sample size of these datasets, please comment on the uncertainty of the results.  How confident are you that the methods actually improve the prediction?  How were the competing models tuned and optimized?

**Experience Assessment:**

I have published in this field for several years.

**Review Assessment: Checking Correctness Of Derivations And Theory:**

I assessed the sensibility of the derivations and theory.

**Review Assessment: Checking Correctness Of Experiments:**

I carefully checked the experiments.

**Review Assessment: Thoroughness In Paper Reading:**

I read the paper thoroughly.

---

> ### Author Response · Authors · 2019-11-14
> **Thank you for reviewing our paper. We have addressed your comments in below. We hope our explanation helps clarifying the unclear points.**
>
> >> First, in the synthetic data the mMLP with the l_QRE loss outperforms the l_quad loss with regards to both E_QRE and E_quad. Why does mMLP/l_QRE outperform on E_quad? As l_quad is actually designed to minimize this error, this is surprising and needs additional explanation.
>
> Note that the evaluation is on the test set and not on the train set. When we use l_quad in mMLP, the generalization is poor on the test set. So it is very much expected from mMLP/l_QRE to outperform on E_QRE, E_quad and in fact any other relevant metric. This highlights that the mMLP/l_QRE has a good generalization on the test set which is really what matters.
>
> >> Additionally, there are two major changes between mMLP and the MLP.  One is the network structure, and the second is the trace being normalized; which change really induces the improvements in the performance? If the l_QRE loss was used in the MLP, would you get similar improvements?
>
> We used l_QRE in MLP in Section 5.1.3. Table 2 shows the result of the evaluation. The result showed that MLP benefits from l_QRE but still under-performs in comparison to the mMLP/l_QRE. That suggests both the loss and the architecture are important.
>
> >> The network structure as a whole needs greater validation. A major difference in (3) is that the weight matrices W_l are applied as a both a left and right multiplication.  Given that the \mathcal{H} operation symmetrizes and normalizes the matrix, a symmetric operation isn't strictly necessary here.  Using both multiplications leads to quadratic properties, which in my experience are less stable in the optimization.  Can the authors validate this structure versus the simpler structure of simply using left multiplications?  Or, in other words, is this weight multiplication structure helpful or do the benefits really come from the \mathcal{H} operation.
>
> What you are suggesting is similar to the experiment that we carried out in Section 5.1.3. There, we compared the architecture of (3) with (18) which is a simpler architecture with only left multiplication. Table 2 summarizes the result of the experiment.
>
> >> I think that the heteroscedastic regression experiments don't evaluate on one of the key issues, which is uncertainty estimation. The trace-1 normalization is highly restrictive, so I...
>
> There is a misunderstanding here. Please take a second look at equation (16) and (17). As you can see the information about the trace is all captured by the scale parameter \eta. That means for the Gaussian case in (16), the Var[y|X] is \eta\Omega where \omega is a trace-1 matrix and \eta includes the trace term. This is stated in the line below equation (16) and below equation (17). So we are indeed learning the full covariance including the trace term. We are basically learning the full covariance as two individual terms: trace term \eta and trace-1 matrix \Omega. The model should capture uncertainties just as expected and trace-1 normalization imposes no restriction since, in addition to the trace-one matrix \omega, we are also learning its normalization \eta.
>
>  >> Also, heteroscedastic regression has a long history in neural networks, dating back to at least Nix and Wiegand in 1994.  The manuscript needs to be updated to...
>
> Thank you for the comment, we will update the related work.
>
> >> Please check Table 5(a), which states that you are only using a small number of training samples.  Also, given the relatively small sample size of these datasets, please comment on the uncertainty of the results.  How confident are you that the methods actually improve the prediction?  How were the competing models tuned and optimized?
>
> Indeed one of the central advantages of the mMLP is that it works well even for the cases with small data size. This was stated in the abstract and empirically validated throughout the paper. For the experiment in Table 5.a, we ran experiments for 10 times and performed a t-test as shown in Fig 5.b. In cases where they are statistical significance (p-value 0.05) we have indicated the results with bold-faced numbers.
>
> >> How were the competing models tuned and optimized?
>
> Table 3 shows the model specifications of the various methods used in the analysis. For the MLP and mMLP methods, we used early stopping on the validation set. For both methods, we used Adam as the optimizer, and considered scheduling of the learning rate. We experimented with various batch sizes, and found that in both cases smaller batch-sizes are preferred though the results are not sensitive to the tuning of this hyperparameter. For the NBCR, the key tuning parameter with the most effect was the truncation. We tried different values ranging from 5 to 20 (step size of 5). In general, due to the need for sampling, the tuning of the NBCR is quite time-consuming. For some cases, one might find better hyperparameter settings for the truncation. However, that would only marginally influence the results at least in our experience.

---

### Official Review · AnonReviewer1 · 2019-10-24
**Official Blind Review #1**

**Rating:** 3

**Review:**

This paper explores the problem of deep heteroskedastic multivariate regression where the goal is to regress over symmetric positive definite matrices; that is, the deep learning model should take as input data points, and produce a conditional covariance matrix as the output. The key challenge in this setting is how to ensure the predicted matrix is positive definite (and thus follows the non-linear geometry of these matrices), how the neural network can be trained for this task, and what loss function can be used for effective training. The paper proposes a neural network with bilinear layers in this regard, and uses the von Neumann divergence as the loss function to regress the predicted covariance against a ground truth SPD matrix. The gradients of the von Neumann divergence are provided for learning via backpropagation. Experiments on several synthetic datasets and small scale datasets are provided, showcasing some benefits.

Pros:
1. The use of von Neumann divergence as a loss for this task is perhaps novel.
2. The use of \alpha-derivatives, while computationally demanding, is perhaps novel in this context as well.

Cons:
1. I do not think the problem setting or the proposed framework is entirely new or is the best choice of its ingredients. Specifically, the idea of using second-order neural networks have been attempted in several prior papers, including the ones the paper cite (such as Ionescu et al. ,2015). Several other papers in this regard are listed below.
[a] Second-order Temporal pooling for action recognition, Cherian and Gould, IJCV, 2018
[b] Deep manifold-to-manifold transforming network, Zhang et al, ICIP, 2018
[c] Statistically motivated second-order pooling, Yu and Salzmann, ECCV, 2018

In comparison to these methods, it is not clear how the proposed setup is novel, or in what way is method better. There are no comparisons to these methods, and thus it is difficult to judge the benefits even empirically.

2. There are also models that predict the mean and covariance matrices directly from the model, such as the works below. The paper should also include and perhaps compare to their datasets.
[d] Deep Inference for Covariance Estimation: Learning Gaussian Noise Models for State Estimation, Liu et al, ICRA, 2018

3. I do not think the use of von Neumann divergence as a loss is the best choice one could have, esp. for a deep neural network learning setting. This divergence includes the matrix logarithm, which is perhaps computationally expensive. This is a problem when using other popular loss/similarity functions on SPD matrices (such as the log-Euclidean metric). Perhaps a better option is to use the Jensen-Bregman log-det divergence, as suggested in [a] above; this divergence is symmetric and also has computationally efficient gradients. It is unclear why the paper decided to use von Neumann.

4. The experiments are not compelling, there are no comparisons to alternative models and the datasets used are small scale. Thus, it is unclear if the design choices in the paper have any strong bearing in the empirical performances.

Overall, the paper makes an attempt at designing neural networks for learning SPD matrices. While, there are some components in the model that are perhaps new, the paper lacks any justifications for their choices, and as such these choices seem inferior to alternatives that have been proposed earlier. Also, the experimental results are not convincing against prior works.



**Experience Assessment:**

I have published in this field for several years.

**Review Assessment: Checking Correctness Of Derivations And Theory:**

I carefully checked the derivations and theory.

**Review Assessment: Checking Correctness Of Experiments:**

I assessed the sensibility of the experiments.

**Review Assessment: Thoroughness In Paper Reading:**

I read the paper thoroughly.

---

> ### Author Response · Authors · 2019-11-14
> **Thank you for reviewing the paper. We hope our explanation helps clarifying the unclear points.**
>
>
> 1- With respect to your comment on the prior papers, we would like to ask you if in all fairness the suggested papers are comparable out of the box or even if they are at all comparable? The fact that our paper shares few keywords including 2nd-order neural network does not mean that they are designed for the same purpose and can be compared against each other. We could of course cite these papers as example of the papers that have used 2nd-order neural network but we do not see how these papers can be compared within the scope of our paper and the problem we are solving.
>
> 2- It is an interesting paper for the sake of comparison, however, the implementation of this work is not available. That being said, the referred paper uses the Cholesky decomposition for preserving the SPD constraint. As discussed in our paper, the Cholesky-based solutions are in general heuristic as they do not consider the non-Euclidean geometry of the SPD matrices. This was evaluated in Section 5.1.2 of our paper.
>
> 3- Please have a look at Figure 1 and Table 1. You can see that the gain in some cases is almost more than 100 times over the quadratic loss. In page 3 and page 4, we have derived a deterministic solution to the derivative of the von-Neumann divergence. We have not explicitly tried the suggested Jensen-Bregman log-det divergence. But we have tried Stein divergence, in our arxiv version of this paper, which is closely related to the log-det divergence and that has resulted in considerably weaker performance in comparison to the von-Neumann divergence. With respect to the computational complexity, in this work we focus on problems with modest dimensionalities. The other point is that there are indeed quite a few other Bregmann divergences.
>
> 4- We think the experiments are compelling enough given that the paper’s main contribution is theoretical. Table 5 lists experiments on 6 real datasets, with various size and dimensionality, and shows comparison with three other methods: multilayer perceptron, Gaussian process, and NBCR.

---

### Decision · Program_Chairs · 2019-12-19

**Decision:**

Reject

**Comment:**

This paper introduces a novel architecture and loss for estimating PSD matrices using neural networks.  There is some theoretical justification for the architecture, and a small-scale but encouraging experiment.

Overall, I think there is a sensible contribution here, but there are so many architectural and computational choices presented together at once that it's hard to tell what the important parts are.

The main problems with this paper are:
1) The scalability of the approach O(N^3)
2) The derivation of the architecture and gradient computations wasn't clear about what choices were available and why.  Several alternative choices were mentioned but not evaluated.  I think the authors also need to improve their understanding of automatic differentiation.  Backprop through eigendecomposition is already available in most autodiff packages.  It was claimed that a certain kind of matrix derivative provided better generalization, which seems like a strong claim to make in general.
3) The experimental setup seemed contrived, except for the heteroskedastic regression experiments, which lacked competitive baselines.  Why were the GP and MLPs homoskedastic?

As a matter of personal preference, I found that having 4 different "H"s differing only in font and capitalization for the network architecture was hard to keep track of.

I agree that R1 had some unjustified comments and R2's review was contentless.  I apologize for these inadequate reviews.